# Consumption of Wild Rice (*Zizania latifolia*) Prevents Metabolic Associated Fatty Liver Disease through the Modulation of the Gut Microbiota in Mice Model

**DOI:** 10.3390/ijms21155375

**Published:** 2020-07-29

**Authors:** Xiao-Dong Hou, Ning Yan, Yong-Mei Du, Hui Liang, Zhong-Feng Zhang, Xiao-Long Yuan

**Affiliations:** 1Tobacco Research Institute of Chinese Academy of Agricultural Sciences, Qingdao 266109, China; houxiaodong@caas.cn (X.-D.H.); yanning0532@163.com (N.Y.); duyongmei0532@163.com (Y.-M.D.); zhangzhongfeng@caas.cn (Z.-F.Z.); 2College of Public Health, Qingdao University, Qingdao 266101, China; qdlianghui@qdu.edu.cn

**Keywords:** gut microbiota, inflammation, insulin resistance, nonalcoholic fatty liver disease, wild rice

## Abstract

Metabolic associated fatty liver disease (MAFLD) due to excess weight and obesity threatens public health worldwide. Gut microbiota dysbiosis contributes to obesity and related diseases. The cholesterol-lowering, anti-inflammatory, and antioxidant effects of wild rice have been reported in several studies; however, whether it has beneficial effects on the gut microbiota is unknown. Here, we show that wild rice reduces body weight, liver steatosis, and low-grade inflammation, and improves insulin resistance in high-fat diet (HFD)-fed mice. High-throughput 16S rRNA pyrosequencing demonstrated that wild rice treatment significantly changed the gut microbiota composition in mice fed an HFD. The richness and diversity of the gut microbiota were notably decreased upon wild rice consumption. Compared with a normal chow diet (NCD), HFD feeding altered 117 operational taxonomic units (OTUs), and wild rice supplementation reversed 90 OTUs to the configuration in the NCD group. Overall, our results suggest that wild rice may be used as a probiotic agent to reverse HFD-induced MAFLD through the modulation of the gut microbiota.

## 1. Introduction

With the improvement of social living standards, the daily diet structure of people has undergone a tremendous change from plant-based foods to high-fat and low-carb carbohydrates. Long-term excessive high-fat intake could induce obesity, which has been linked with major degenerative diseases and the development of metabolic (dysfunction) associated fatty liver disease (MAFLD), formerly known as non-alcoholic fatty liver disease (NAFLD) [1,2]. MAFLD affects about 25% of the global population and is associated with metabolic derangements such as visceral obesity, increased fasting blood glucose, dyslipidemia, and hypertension [2]. Within the global MAFLD population, around 60% and 40% of the population was classified as obese and non-obese, respectively [3]. Moreover, the incidence of MAFLD varies worldwide, and is relatively high in developed countries such as Europe and the United States [4,5]. In developing countries, the incidence of the disease is about 15%, but in obese patients, the incidence is as high as 50% or more [4]. It has been shown that MAFLD is not associated with liver-related disease, but increasing research has demonstrated that MAFLD is linked with various chronic diseases [6]. For example, MAFLD can increase the risk of hypertension, type 2 diabetes mellitus, oxidative stress, chronic kidney disease, etc., which all threaten human health [6,7,8]. MAFLD and its associated metabolic syndrome have become a global concern.

Growing evidence indicates that MAFLD and metabolic syndrome can influence the gut microbiota compositions [9]. The gut microbiota is a diverse ecosystem composed of bacteria, archaea, fungi, and viruses. Existing studies have found that patients with obese and non-obese MAFLD have an imbalance of gut microbiota [10,11,12]. For example, there was an increased abundance of the species *Dorea* and a reduction in the relative abundance of a number of species, including *Marvinbryantia* and the *Cbristensellenaceae* R7 group in patients with non-obese MAFLD [13]. Interestingly, mice fed the cholesterol-rich diet were observed to have an increased relative abundance of *Bacteroidetes* and a decrease in *Firmicutes* compared with those fed with high sucrose, suggesting a clearly differential profile of gut microbiota between obese and non-obese subjects with MAFLD [13]. Recent reports have shown that the gut microbiota may promote the progress of MAFLD through the intestinal–hepatic axis pathway [14]. More specifically, increased intestinal permeability and bacterial translocation can allow intestinal microbes and metabolites to reach the liver, thereby affecting bile acid metabolism, and promoting intestinal motility disorders and systemic inflammation [14]. Furthermore, the gut microbiota can contribute to the occurrence and development of MAFLD by regulating energy balance, increasing the synthesis of short chain fatty acids (SCAs) and triglycerides, regulating choline metabolism and bile acid balance, producing endogenous ethanol, bacteria-derived toxins, and promoting the release of pro-inflammatory cytokines in the liver [15,16,17].

Despite tremendous progress in the research on the pathogenesis of MAFLD, an effective therapeutic method is still being actively explored. A large number of studies have shown that changing the diet structure can help patients with non-alcoholic fatty liver disease improve their condition. Wild rice (WR) is a genus of grasses (*Zizania*), but not a type of rice. There are four species of WR belonging to the genus *Zizania*. *Zizania aquatica* L., *Zizania palustris* L., and *Zizania texana* Hitche are native to North America, whereas *Zizania latifolia* (Griseb) Turcz is native to China, Japan, and Vietnam [18]. Wild rice is now considered a functional food, and is currently available in grocery stores in North America because of several of its characters, such as low contents in cholesterol and its abilities in anti-inflammatory and antioxidant aspects [18,19,20]. Because WR has a chewy dark outer layer and soft inner grain after cooking, it has been used as a component of salads or steamed mixed grains. Numerous reports have shown that WR has a higher content of protein; dietary fiber; vitamin B1, B2, E; and minerals than common white rice [20,21,22]. Moreover, the contents of phenolic acids and flavonoids in wild rice are significantly higher than that in white rice [23,24]. Plenty of research suggests proanthocyanidins plays a key role in changing the composition of the gut microbiota based on in vitro and in vivo models and the bioactivities of their metabolites [25,26,27]. However, it has not been reported that consumption of WR could prevent MAFLD via the modulation of the gut microbiota. Therefore, we assessed the effect of WR on nonalcoholic fatty liver disease induced by a high-fat diet. Furthermore, we found that WR causes substantial changes in the gut microbiota composition, suggesting WR has attenuated many of the adverse health consequences associated with an HFD, and it modulates the gut microbiota in a positive way.

## 2. Results

### 2.1. General Nutrition Composition

The protein contents of the wild rice samples (15.6 ± 0.21 g/100 g) were markedly higher than that of the white rice (10.19 ± 0.29 g/100 g) and red rice (11.78 ± 0.11 g/100 g; Table 1). The average fat and ash content of the wild rice was similar to red rice, and was 1.67 and 4.36 times as high as that of white rice, respectively. The moisture was about 10% in all of the rice samples, and less than that in white rice. The sodium content, dietary fiber, resistant starch, and total phenolic in wild rice exceeded those of white rice and red rice. Wild rice is a caryopsis with a seed coat that contains a lot of minerals and dietary fiber. Hence, wild rice is considered to be a possible food for people with diabetes.

### 2.2. WR Attenuates Features of Obesity in HFD-Fed Mice

Compared with the NCD mice, the HFD-fed mice gained more weight (Figure 1A) and developed hallmark features of hyperlipidemia, including serum lipid profiles (increased serum TG (Figure 1B), TC (Figure 1C), LDL (Figure 1D), and decreased serum HDL (Figure 1E)), hepatic lipid profiles (liver TG (Figure 1F), TC (Figure 1G), LDL (Figure 1H), and decreased liver HDL (Figure 1I)), increased impaired glucose clearance in the oral glucose tolerance test (OGTT; Figure 1J,K), and insulin resistance (Figure 1L,M). Treatments with WR significantly attenuated HFD-induced weight gain (Figure 1A). The attenuated weight gain was associated with modestly reduced serum and liver TG (Figure 1B,F), TC (Figure 1C,G), and LDL (Figure 1D,H), as well as increased serum and liver HDL (Figure 1E,I). In addition, dietary supplementation with WR significantly increased the clearance of a glucose bolus during the OGTT (Figure 1J,K) and improved HFD-induced insulin resistance (Figure 1L,M) Thus, WR enhanced the glucose–insulin homeostasis by attenuating HFD-induced MAFLD.

### 2.3. WR Prevents HFD-Induced Liver Steatosis, Oxidative Stress, and Systemic Low-Grade Inflammation

Next, the liver steatosis was evaluated by HE staining and transmission electron microscopy. The HFD group showed representative characteristics of liver steatosis (Figure 2A,B), such as a large number of lipid droplet vacuoles, irregular arrangement of cells, and cytoplasm rarefaction, an effect that was ameliorated by the WR treatment (Figure 2A,B). Previous studies have shown that obesity and liver steatosis are associated with liver oxidative stress. We evaluated the effects of WR on liver oxidative stress by examining the level of SOD, GSH-Px, and MDA. The SOD (Figure 2C) and GSH-Px (Figure 2D) levels in the HFD mice were significantly lower than in the NCD mice, and the MDA content (Figure 2E) was significantly higher than in the control group. As expected, an HFD supplemented with WR significantly increased the SOD and GSH-Px content (Figure 2C,D), and reduced the MDA content compared with the HFD mice (Figure 2E). Previous studies have shown that liver oxidative stress is associated with low-grade chronic inflammation. We investigated the effects of WR on the serum inflammatory cytokine (IL-1, IL-6, and IL-8) levels. WR treatment reduced systemic low-grade inflammation in HFD-fed obese mice (Figure 2F–H). Because NF-κB is a key player in inflammatory reactions, HFD feeding significantly enhanced the mRNA and protein levels of NF-κB (Figure 2I,K), and reduced the mRNA levels of IκB-α (Figure 2J,K) in the liver, whereas WR supplementation restored their levels (Figure 2I–K).

### 2.4. WR Alters the Gut Microbiota Composition in HFD-Induced Mice

To investigate the effect of WR on the gut microbiota composition in HFD-induced mice, we amplified and sequenced the V3 + V4 region of the 16S rRNA of each sample. Finally, we obtained 1188756 clean tags, which were used for the following analysis (Appendix A). The alpha diversity analysis showed that the HFD-fed mice treated with WR showed significantly reductions in the richness and diversities of their gut microbiota based on the numbers of OTUs (Figure 3A), rarefaction curves (Figure 3B), Chao1 curves (Figure 3C) and index of Chao1 (Figure 3D), index of Observed species (Figure 3E), index of PD whole tree (Figure 3F), and index of goods coverage (Figure 3G). Furthermore, we also found that HFD feeding significantly decreased the *Bacteroidetes* abundance, while it increased the abundance of *Firmicutes*. In contrast, WR recovered these levels and affected the ratio of *Firmicutes* to *Bacteroidetes* (Figure 3H). In addition, we found that the results of the genus level and phylum level analysis were consistent. For example, WR significantly increased the relative abundance of Lactobacillus in HFD-fed mice (Figure 3I). In addition, according to the analysis results of NMDS, PCoA, and PCA, the structure of the WR treatment group was similar to that of NCD group when comparing these indexes with the HFD group. (Figure 3J–L).

### 2.5. WR Modulates the Key Phylotypes of Gut Microbiota in HFD-Fed Mice

In order to explore the microbiology communities induced by WR, we analyzed their effective sequences via redundancy analysis (RDA). In total, 136 predictive OTUs were identified (Figure 4A,B and Appendix A). Among these OTUs, HFD feeding altered 117 OTUs (50 increased and 67 decreased) compared with the NCD group. However, the treatment of WR could generate 90 variable OUTs, among which 70 increased and 20 decreased. Furthermore, we found that 71 OTUs showed reverse correction in the HFD-fed and NCD groups. For example, *Ruminococcaceae*, *Desulfovibrionaceae*, *Bacteroides*, and *Porphyromonadaceae* have been negatively correlated with WR. We compared the gut microbiota of different treatments using LEfSe to identify the specific bacterial taxa associated with WR treatment. The LEfSe analysis showed that WR increased *Lactobacillus* (Figure 4 and Figure 5A), which is a probiotic that helps reduce obesity, diabetes, and inflammation caused by HFD [28,29,30], while it decreased the negative microorganisms such as *Prevotella*, *Bacteroides,* and *Staphylococcus* (Figure 4 and Figure 5B–D). What is more, we found that some known bacteria (such as *Prevotella*, *Bacteroides,* and *Staphylococcus*) with negative effects were observed with decreased trends after treatment in the HFD-fed mice with WR (Figure 4 and Figure 5B–D). These results demonstrate that WR treatment reverses some of the changes in the gut microbiota composition induced by a HFD.

## 3. Discussion

The gut microbiota is closely correlated with MAFLD and inflammation. Although several studies have shown that wild rice has anti-MAFLD and anti-inflammatory activities in different animal models [18,31,32,33], the effects of wild rice on the gut microbiota have not yet been investigated. Herein, we provide evidence that wild rice oral administration for 11 weeks has a protective effect against dietary-induced MAFLD, insulin resistance, liver steatosis, and low-grade inflammation via the modulation of the gut microbiota. As already mentioned, the gut microbiota of MAFLD humans and rodent models is highly associated with decreased abundances of *Bacteroidetes* and increased abundances of *Firmicutes* [34,35,36]. Our results show that wild rice supplementation significantly increased the abundance of *Bacteroidetes* and decreased the abundance of *Firmicutes* in HFD-fed mice (Figure 3E,F). The effects could be attributed to the high dietary fiber, resistant starch, and polyphenol content of WD. In addition, the abundance of Gram-negative bacteria, such as *Prevotellaceae*, was increased in the intestinal ecosystem of HFD mice and decreased in WR mice. These effects may be due to the total phenolic in wild rice. Many studies have revealed the effects of proanthocyanidins from different sources against gut microbiota, including *Prevotella* [27,37]. Numerous reports revealed that *Prevotellaceae* and the included genera are associated with low-grade inflammation [38,39,40].

Additionally, our results demonstrate that wild rice significantly enhanced the level of *Lactobacillus* (Figure 3I and Figure 4, Appendix A). Preliminary reports demonstrated HFD, to a greater or lesser degree, decreased the relative abundance of *Lactobacillus*, a probiotic belonging to the *Lactobacillaceae* family, which is associated with nutrient metabolism [41]. In detail, Khare et al. explored the effect of cinnamaldehyde in HFD mice, and found that HFD feeding can decrease the abundance of *Lactobacillus* species [42]. Furthermore, the abundance of *Lactobacillus* showed a decreasing tendency when investigating the effect of α-cyclodextrin on the gut microbiota in HFD mice [43]. Similarly, the microbiological analysis of the gut microbiotas in HFD mice showed that the abundance of Lactobacillus decreased, while it increased when treated with the pandanus tectorius fruit extract and isomalto-oligosaccharides [44,45]. Moreover, *Bacteroides*, a genus in the family *Bacteroidetes*, was associated with increases in the acetic acid and propionic acid levels in the gut, and its abundance was also restored by wild rice in the HFD-fed mice (Figure 4), which is consistent with previous findings [46,47]. *Prevotella* predominantly stimulates cells to produce IL-1, IL-6, and IL-8, resulting in metabolic disorders and low-grade systemic inflammation [48]. The level of *Prevotella* was reduced by wild rice in HFD-fed mice in the present study (Figure 4 and Figure 5B).

Recent studies have shown an interrelation between oxidative stress and low-grade inflammation in humans and rodents [49]. Our research demonstrated that the antioxidant status of HFD-fed mice was significantly improved by treatement with WR, characterized by an increased content of MDA and decreased activities of GSH-Px and SOD (Figure 2C–E), which is consistent with previous research [18]. It has also been reported that there is a relationship between obesity and low-grade inflammation. Our study showed similar results with a previous study, that the accumulation of hepatic lipids induced by HFD was significantly reduced in the treatment group (Figure 1F–I and Figure 2A,B) [18]. In addition to the changes in the physiological indicators, it has been shown that the NF-κB signaling pathway participated in the processes of inflammation and diabetes [50]. In the WR-treated groups, all markers of systemic low-grade inflammation, such as TNF-α, IL-1, IL-6, IL-8, and NF-κB, exhibited downregulation, whereas IκB-α was induced (Figure 1M and Figure 2F–K). Additionally, the circulating TG, TC, blood glucose, and insulin levels increased in the HFD-fed mice. Wild rice, on the other hand, protected against increased insulin resistance in obese mice (Figure 1). The purpose of this study was to unveil the potential influence of the wild rice on the gut microbiota and the NF-κB signaling pathway. Further studies are still needed in order to elucidate its mechanism.

## 4. Materials and Methods

### 4.1. Diet Preparation

Whole grains of wild rice (Z. *latifolia*) were hand-harvested from Baimahu Village, Qianfeng Town, Jinhu County, Huai’an City, Jiangsu Province, China (33°11′9″ N; 119°9′37″ E) on 20 September 2017. After air drying, full and plump seeds were picked and ground to a fine powder in a mechanical grinder, sieved through a 0.45 mm sifter, and maintained in a desiccator until use. The seeds and experimental diets were sterilized by γ irradiation at 10 kGy. The normal chow diet (NCD) was purchased from Beijing HFK Bioscience Co., Ltd. (Beijing, China).

### 4.2. Proximate Composition Analysis

The proximate compositions of the crude protein, total lipid, moisture, total ash, sodium (Na), dietary fiber, and t resistant starch of all of the samples were analyzed according to the Chinese National Standards, namely: GB 5009.5–2016, GB 5009.6–2016, GB 5009.3-2016, GB 5009.4-2016, GB 5009.91-2016, GB/Z 21922-2008 2.2.8, GB28050-2011, GB 5009.88-2014, and NY/T 2638-2014, respectively.

### 4.3. Animal Trial

Ten-week-old male C57BL/6J mice were obtained from Shandong Experimental Animal Center (Jinan, China). All of the mice were fed an identical diet of normal chow during the first two weeks. After this period, the mice were randomly divided into four dietary groups (six mice per cage, two cages per group) in a room in a room at 25 ± 2 °C and 55 ± 5% relative humidity with controlled lighting (12:12 h light/dark cycle), with free access to water and the appropriate diets. The NCD group received the normal rodent chow, the HFD group received the high-fat diet (normal chow containing 10% lard), and the wild rice diet groups received normal chow containing 10% lard + 10% wild rice or 20% wild rice for 11 weeks. The compositions of the experimental diets are shown in Appendix A. The food intake was recorded daily, and the body weight was recorded every week. All efforts were made to reduce animal suffering. Experimental procedures were performed according to the guidelines of the Ethics and Animal Welfare Committee of Shandong Academy of Medical Sciences (Jinan, China, SCXK-20140007). All of the sections of this report adhere to the ARRIVE Guidelines for reporting animal research [51].

### 4.4. Biochemical Analysis

Blood samples were collected from fasted mice at the end of the eleventh week. The serum was separated by centrifuging the samples at 2000× *g* for 10 min at 4 °C, and was then stored in sterile Eppendorf tubes at 80 °C. The livers were carefully removed, cut into several sections, and then stored at −80 °C after snap-freezing in liquid nitrogen. A total of 0.1 g liver tissue was homogenized with 0.9 mL of saline using a homogenizer (Biospec Products, Bartlesville, OK, USA). The liver homogenate was centrifuged at 1800× *g* at 480 °C for 15 min, and the supernatant was subjected to further analysis. Serum parameters including total cholesterol (TC), triglycerides (TG), low density lipoprotein (LDL), and high density lipoprotein (HDL) levels, as well as liver parameters, including TC, TG, LDL, HDL, interleukin-1(IL-1), IL-6, IL-8, rat tumor necrosis factor alpha (TNF-α), insulin, superoxide dismutase (SOD), malondialdehyde (MDA), and glutathione peroxidase (GSH-px) levels, were determined using analytical reagent kits (Nanjing Jiancheng Bioengineering Institute, Nanjing, Jiangsu, China). A microplate reader (Multiskan GO, Thermo Fisher Scientific, Waltham, MA, USA) was used to measure the kinetic UV. Each assay was performed at least three times.

### 4.5. Oral Glucose Tolerance Tests

For the oral glucose tolerance tests, during the eighth week of the experiment, the mice were fasted for 12 h before receiving an oral administration of D-glucose (1.5 g/kg). Blood samples were taken from the tail vein at 30 and 120 min following glucose administration, and the blood glucose was measured using a glucometer (Johnson and Johnson, Shanghai, China).

### 4.6. Liver Histopathology

Approximately 8-μm-thick frozen liver sections were prepared to measure the accumulation of hepatic fat. The histopathological changes in the frozen liver sections of the mice were examined using HE staining, as previously described [52].

### 4.7. Liver Ultrastructure Examination

The liver ultrastructure was investigated using transmission electron microscopy (TEM), as described previously [53]. Briefly, livers were collected, as mentioned above, and fixed with 2% glutaraldehyde (containing 0.1 M sucrose and 0.2 M sodium cacodylate) overnight at 4 °C, followed by 10 g/L osmium tetroxide. The samples were dehydrated and embedded in epoxy resin. Ultra-thin sections were observed using an H700 TEM (Hitachi, Tokyo, Japan).

### 4.8. Western Blot Analysis

The liver protein was extracted using radio immunoprecipitation assay (RIPA) lysis. The protein amount was measured using a BCA protein assay reagent from Pierce (Rockford, IL, USA). The protein samples were separated by 10% SDS-PAGE gel and transferred onto Immobilon^®^-P transfer membranes (Sigma–Aldrich, St. Louis, MO, USA). After blocking with 8% nonfat milk, the membranes were first incubated with primary antibodies against nuclear factor (NF)-κB (1:2000; Cell Signaling Technology, Danvers, MA, USA), IκB-α (1:1000; Cell Signaling Technology), and β-actin (1:6000; Cell Signaling Technology) at 4 °C for 12 h, and then with the secondary antibody for 1 h, as described previously [54]. The intensities of the protein bands were quantified with Image J software (http://imagej.nih.gov/ij/) and the values normalized to β-actin.

### 4.9. Gene Expression Analysis

The tissue total RNA was isolated with an RNAprep Pure tissue kit from Tiangen Biotech Co., Ltd. (Beijing, China). For real-time PCR analysis, reverse transcription was performed with oligdT-18 and M-MLV transcriptase from Promega (Madison, WI, USA). Real-time PCR was performed using SYBR Green qPCR Super Mix (Invitrogen, San Diego, CA, USA) on an ABI 7500 real-time PCR system (Applied Biosystems, Foster City, CA, USA), according to the manufacturer’s instructions, as described previously. The values were normalized against the control glyceraldehyde-3- phosphate dehydrogenase (GAPDH). The sequences of the real-time PCR primers used are listed in Appendix A.

### 4.10. Gut Microbiota Analysis

On the last day of the animal trial for the gut microbial analysis, we collected the fresh fecal samples and then extracted the total DNA following the manufacturer’s recommendations. After measuring the concentration of DNA using a Nanodrop 8000 spectrophotometer (Thermo Fisher Scientific), the DNA sample was diluted with sterile water to a concentration of 20 ng/L. The sequencing library of the bacterial 16S rRNA gene V3-V4 region in the DNA samples was constructed by PCR using bacterial primers (Appendix A), according to the modified guide provided by Illumina (Part # 15044223 Rev. B). The raw data were firstly merged with FLASH [55] and then separated based on their unique sequencing barcodes. All high-quality sequencing sequences were merged, and the effective sequences were obtained after removing the exact same sequences. The effective sequences were clustered into operational taxonomic units (OTUs) according to their sequence similarities (above 97%) [56]. Based on the Quantitative Insight IntoMicrobial Ecology, v. 1.8.0 (QIIME) platform, the alpha diversity analysis and beta-diversity analysis were analyzed [57,58,59]. The total OTU number, Shannon diversity index, and Faith’s phylogenetic diversity (PD whole tree) were used to indicate the alpha diversity of each sample, and the Mann–Whitney U test was used in GraphPad Prism 7 software to analyze each group. Multi values such as the OTU rank curves, Shannon index, calculated indexes of Chao1, PD whole tree index, and goods coverage were used for calculating their alpha diversities. Meanwhile, we assessed the beta diversities based on nonmetric multidimensional scaling (NMDS), uniFrac distance-based principal coordinate analysis (PCoA), and principal component analysis (PCA), together with RDA-identified key OTUs.

The total OTU number, Shannon diversity index, and Faith’s phylogenetic diversity (PD Whole tree) were used to indicate the alpha diversity of each sample, and the Mann–Whitney U test was used in GraphPad Prism 7 software to analyze in each group the significance of the difference. Beta diversity analysis was mainly based on the principal coordinate analysis (PCoA), based on the Bray–Curtis distance and the normalized sample-OTU abundance matrix. The obtained PCoA score map can visually show the small groups in the 10th week of intervention. The dynamic trajectory of the overall structure of the rat intestinal flora [57,58,59]. The LDA effect size (LEfSe) analysis was used to perform the quantitative analysis of the biomarkers for each group [60].

### 4.11. Statistical Analysis

The results were presented as means ± the standard error of the mean (SEM). Statistical analysis was performed using SPSS, version 20 (IBM, Armonk, NY, USA). The differences between groups were statistically analyzed using ANOVA, followed by LSD multiple comparison tests and unpaired t tests, and were considered statistically significant at a level of *p* < 0.05.

### 4.12. Accession Number

The sequence information in this paper was submitted to the GenBank Sequence Read Archive database under accession number SRP020353.

## 5. Conclusions

In conclusion, our results show that wild rice treatment modulates the gut microbiota, promoting a decrease in the ratio of *Firmicutes* to *Bacteroidetes,* and increasing the relative abundance of *Lactobacillus*, while bringing the abundances of *Prevotella*, *Alloprevotella*, *Anaerovibrio*, and *Staphylococcus* back to normal levels, thereby providing beneficial effects against obesity, insulin resistance, liver steatosis, and low-grade inflammation to alter MAFLD in HFD-fed mice.

## Figures and Tables

**Figure 1 ijms-21-05375-f001:**
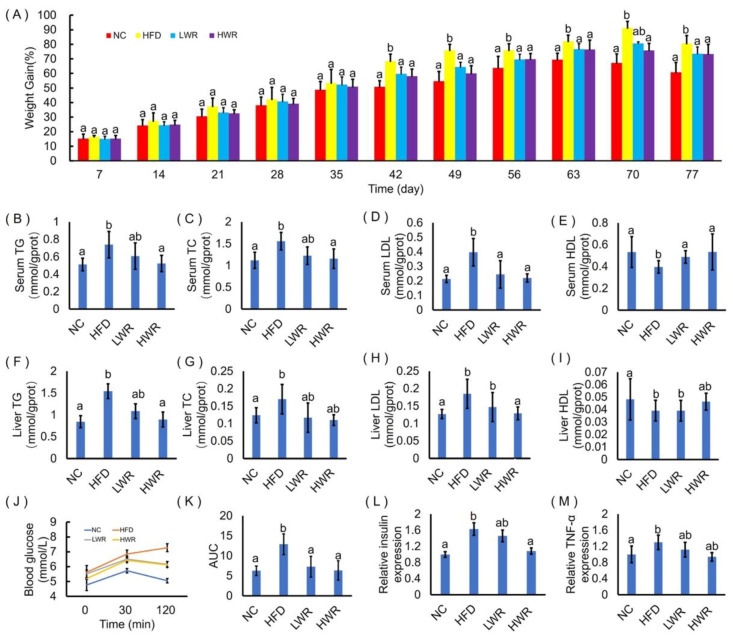
Wild rice attenuated high-fat diet (HFD)-induced obesity and impaired glucose–insulin homeostasis. (**A**) Body weight gain as a percentage of baseline weight for each mouse. (**B**) Serum concentrations of triacylglycerol (TG). (**C**) Serum concentrations of total cholesterol (TC). (**D**) Serum concentrations of low-density lipoprotein (LDL). (**E**) Serum concentrations of high-density lipoprotein (HDL). (**F**) Liver concentrations of triacylglycerol (TG). (**G**) Liver concentrations of total cholesterol (TC). (**H**) Liver concentrations of low-density lipoprotein (LDL). (**I**) Liver concentrations of high-density lipoprotein (HDL). (**J**) Curve of oral glucose tolerance test (OGTT). (**K**) Areas under the curve (AUC) of OGTT. (**L**) Serum relative expression of insulin (INS). (**M**) Liver relative expression of tumor necrosis factor-α (TNF-α). NC presents normal chow diet group. HFD presents high-fat diet group. LWR presents low dose wild rice diet group. HWR present high dose wild rice diet group. a,ab,b: The values of each group with the same letters are not significantly different in the analysis of variance followed by Tukey’s post-hoc test (*n* = 8 mice per group).

**Figure 2 ijms-21-05375-f002:**
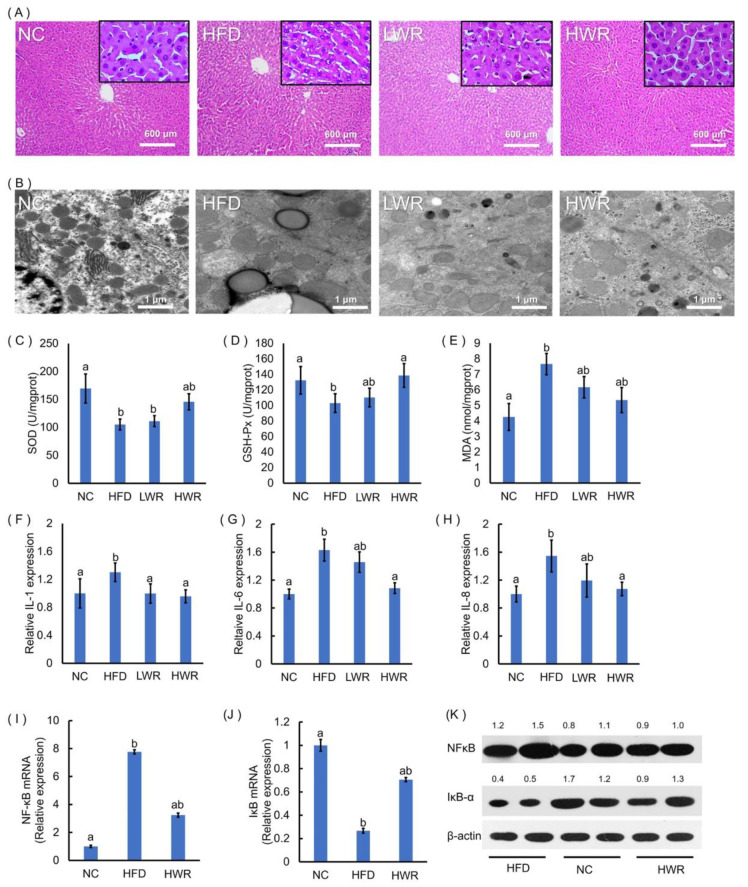
Wild rice prevents HFD-induced adipocyte hypertrophy, hepatic steatosis, and systemic inflammation. (**A**) Hematoxylin and eosin (HE)-stained liver sections. (**B**) Ultrastructural changes in the liver were detected via TEM. The (**C**) liver superoxide dismutase (SOD), (**D**) GSH-Px, and (**E**) malondialdehyde (MDA) were measured using commercial assay kits. The (**F**) interleukin (IL)-1, (**G**) IL-6, and (**H**) IL-8 concentrations were measured using ELISA. Gene expression levels of (**I**) NFκB and (**J**) IκB-α in the liver were measured by qRT-PCR. (**K**) Liver NFκB and IκB-α protein production was examined by Western blot, and relative protein levels were normalized with β-actin (*n* = 3). (**C**–**J**) a,ab,b: The values of each group with the same letters are not significantly different in the analysis of variance followed by Tukey’s post-hoc test.

**Figure 3 ijms-21-05375-f003:**
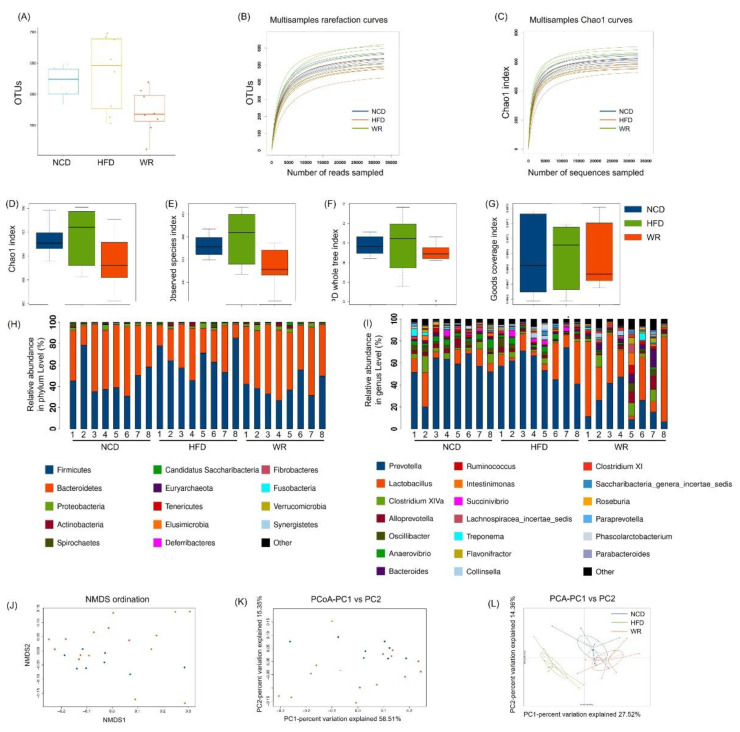
Comparison of the alpha diversities, species richness, and structures of the gut microbiota between wild rice and HFE-fed mice. (**A**) The numbers of OTUs. (**B**) Rarefaction curves and (**C**) Chao1 curves. (**D**–**G**) Index of the Chao1 index, observed species, phylogenetic diversity (PD) whole tree, and goods coverage of each group, respectively. All of the values are presented as mean ± SE (*n* = 5). Relative abundances at the level of the (**H**) phylum and (**I**) genus level. (**J**–**L**) NMDS, PCoA, and PCA of each sample. NCD presents normal chow diet group. HFD presents high-fat diet group. WR presents wild rice diet group, respectively.

**Figure 4 ijms-21-05375-f004:**
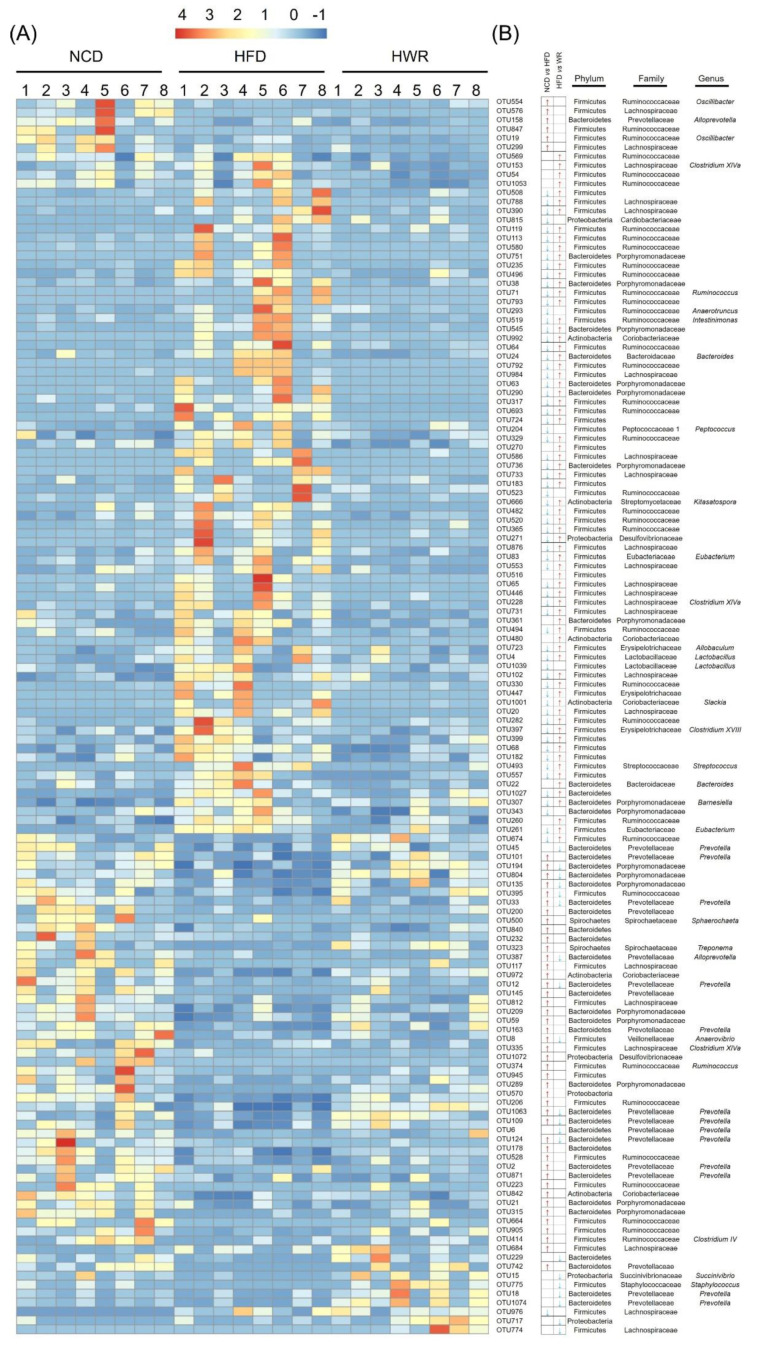
Heatmap and representative bacterial statistics of redundancy analysis (RDA)-identified OTUs and LefSe among differentially taxons in the normal chow diet (NCD) group and experimental groups. (**A**) Heat map of the relative abundance of RDA-identified key 136 OTUs. (**B**) Statistics of representative bacterial taxons at the genus, family, and phylum levels based on OTUs. The red and blue arrows represent the increased or decreased OTUs in the normal chow diet (NCD) and HWR groups relative to the HFD-fed group (*p* < 0.01).

**Figure 5 ijms-21-05375-f005:**
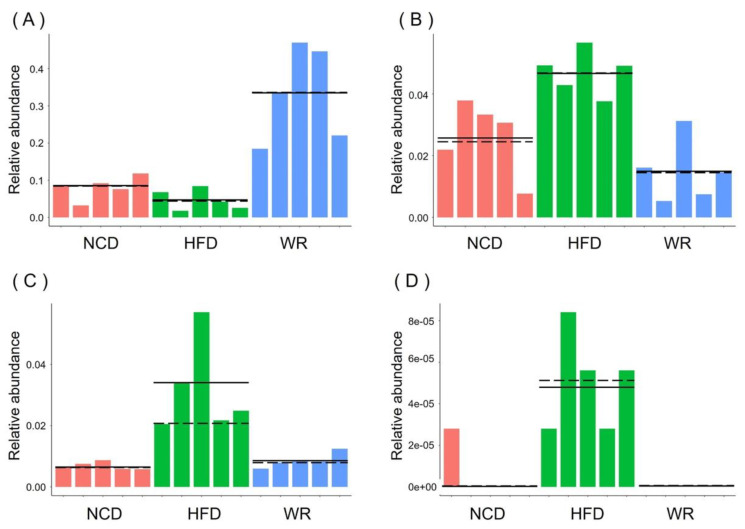
The relative abundance of (**A**) *Lactobacillus*, (**B**) *Prevotella*, (**C**) *Anaerovibrio*, and (**D**) *Staphylococcus* obtained in fecal microbiota from the LefSe results. Solid and dashed lines indicate the mean and median, respectively.

**Table 1 ijms-21-05375-t001:** Comparison of general nutrition composition among wild rice, white rice, and red rice (mean ± standard deviation (SD)).

Composition	White Rice (g/100 g)	Red Rice (g/100 g)	Wild Rice (g/100 g)
Crude protein	10.19 ± 0.29 ^a^	11.78 ± 0.11 ^a^	15.6 ± 0.21 ^b^
Total fat	0.67 ± 0.01 ^a^	1.36 ± 0.05 ^b^	1.12 ± 0.01 ^b^
Moisture	11.76 ± 0.10 ^a^	10.61 ± 0.17 ^a^	10.16 ± 0.12 ^a^
Total ash	0.30 ± 0.02 ^a^	1.38 ± 0.02 ^b^	1.31 ± 0.25 ^b^
Sodium	11.8 ± 0.87 ^a^	84.5 ± 4.55 ^b^	5.21 ± 0.61 ^c^
Dietary fiber	0.42 ± 0.01 ^a^	2.68 ± 0.01 ^b^	6.83 ± 0.11 ^c^
Resistant starch	1.41 ± 0.04	0.95 ± 0.01	10.87 ± 0.15
Total phenolic [25]	1.30 ± 0.00 ^a^	1.40 ± 0.00 ^a^	2.10 ± 0.00 ^b^

^a,b,c^: The values of each group with the same letters are not significantly different in the analysis of variance followed by Tukey’s post-hoc test (*n* = 5 per group).

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
