# Peer review of "Consumption of Wild Rice (Zizania latifolia) Prevents Metabolic Associated Fatty Liver Disease through the Modulation of the Gut Microbiota in Mice Model"

_ijms, 2020, doi:10.3390/ijms21155375_

Round 1
Reviewer 1 Report
This is an interesting study that provides a new information on the effect of wild rice on the gut microbiota and diet-induced fatty liver disease, however there are some concerns that need to be considered.
1. There is lack of consideration of fatty liver disease in non-obese subjects, though recent studies suggest that 40% of the global fatty liver disease population are non-obese (Lancet Gastroenterol Hepatol. 2020. PMID: 32413340), with clearly differential profile of gut microbiota between obese and non-obese subjects with fatty liver disease (Hepatology . 2020 Apr;71(4):1213-1227. doi: 10.1002/hep.30908. PMID:31442319). This omission needs to be corrected.
2. According to recent international consensus, the nomenclature of fatty liver disease has been updated from NAFLD to MAFLD (Gastroenterology. 2020 May;158(7):1999-2014.e1 J Hepatol. 2020 Apr 8. pii: S0168-8278(20)30201-4). Authors need to update the title and the entire manuscript accordingly.
Author Response
Response to reviewers:
Reviewer 1
Comments and Suggestions for Authors
1. There is lack of consideration of fatty liver disease in non-obese subjects, though recent studies suggest that 40% of the global fatty liver disease population are non-obese (Lancet Gastroenterol Hepatol. 2020. PMID: 32413340), with clearly differential profile of gut microbiota between obese and non-obese subjects with fatty liver disease (Hepatology. 2020 Apr;71(4):1213-1227. doi: 10.1002/hep.30908. PMID:31442319). This omission needs to be corrected.
Response: Thanks for your suggestion. We have revised this in the introduction section according to your suggestions. Detail information can be found in the manuscript.
2. According to recent international consensus, the nomenclature of fatty liver disease has been updated from NAFLD to MAFLD (Gastroenterology. 2020 May;158(7):1999-2014.e1 J Hepatol. 2020 Apr 8. pii: S0168-8278(20)30201-4). Authors need to update the title and the entire manuscript accordingly.
Response: Thanks for your suggestion. We have updated from NAFLD to MAFLD in our manuscript according to your suggestions.
Reviewer 2 Report
In the manuscript entitled “Consumption of Wild Rice (Zizania latifolia) Prevents Nonalcoholic Fatty Liver Disease Through Modulation of Gut Microbiota in Mice Model” Xiao-Dong Hou et al have performed an in-depth effect of Wild rice on the modulation of gut microbiota and associated inflammation, anti-oxidant response etc in high-fat diet conditions. The data is very well represented and the results of effect of WR on positive bacteria, lactobacillus versus its negative effect on bad microorganisms is particularly striking. I have just one small comment below for figure Figure 5A:
The decrease in lactobacillus content in HFD model is small compared to NCD and there is some variation between animals. Is there any previous literature showing similar effects of HFD on lactobacillus in vivo? I know that differential environments may affect how microbiota behave, but if there is any literature, it should be discussed in the discussion section.
Author Response
Response to reviewers:
Reviewer 2
Comments and Suggestions for Authors
In the manuscript entitled “Consumption of Wild Rice (Zizania latifolia) Prevents Nonalcoholic Fatty Liver Disease Through Modulation of Gut Microbiota in Mice Model” Xiao-Dong Hou et al have performed an in-depth effect of Wild rice on the modulation of gut microbiota and associated inflammation, anti-oxidant response etc in high-fat diet conditions. The data is very well represented and the results of effect of WR on positive bacteria, lactobacillus versus its negative effect on bad microorganisms is particularly striking. I have just one small comment below for figure Figure 5A:
The decrease in lactobacillus content in HFD model is small compared to NCD and there is some variation between animals. Is there any previous literature showing similar effects of HFD on lactobacillus in vivo? I know that differential environments may affect how microbiota behave, but if there is any literature, it should be discussed in the discussion section.
Response: Thank you very much for your evaluation and comments on our manuscript. According to your suggestion, we read several related literatures and added some sentences in the discussion section. Detail information can be found in the manuscript.